# Strength–Durability Correlation of Osteosynthesis Devices Made by 3D Layer Manufacturing

**DOI:** 10.3390/ma12030436

**Published:** 2019-01-31

**Authors:** Yoshimitsu Okazaki, Emiko Gotoh, Jun Mori

**Affiliations:** 1Department of Life Science and Biotechnology, National Institute of Advanced Industrial Science and Technology, 1-1 Higashi 1-chome, Tsukuba, Ibaraki 305-8566, Japan; 2Planning and Administration Department, National Institute of Technology and Evaluation, 2-49-10, Nishihara, Shibuya-ku, Tokyo 151-0066, Japan; goto-emiko@nite.go.jp; 3Representative Director, Instron Japan Company Limited, 1-8-9 Miyamaedaira, Miyamae-ku, Kawasaki-shi, Kanagawa 216-0006, Japan; Jun_Mori@Instron.com

**Keywords:** osteosynthesis devices, mechanical performance, bending test, bending strength, bending stiffness, durability, microstructure

## Abstract

To develop orthopedic implants that are optimized for each patient’s needs or skeletal structure (custom-made implants), evaluations of the bending strength, bending stiffness, and durability of various types of conventional osteosynthesis devices have become important. Four-point bending tests and compression bending tests of osteosynthesis devices (bone plates, intramedullary nail rods, spinal rods, compression hip screws (CHSs), short femoral nails, and metaphyseal plates) were carried out to measure their bending stiffness, bending strength, and durability. The bending stiffness of bone plates, intramedullary nails, spinal rods, CHSs, short femoral nails, and metaphyseal plates increased with increasing bending strength. The durability limit of various types of osteosynthesis devices linearly increased with increasing bending strength. The relationship (durability limit at 10^6^ cycles) = 0.67 × (bending strength) (N·m) (R^2^ = 0.85) was obtained by regression. The relationship for the highly biocompatible Ti-15Zr-4Nb-4Ta alloy was also linear. The mechanical strength and ductility of specimens that were cut from various osteosynthesis devices were excellent and their microstructures consisted of fine structures, which were considered to be related to the excellent durability. These results are expected to be useful for the development of implants suitable for the skeletal structure of patients using three-dimensional (3D) layer manufacturing technologies.

## 1. Introduction

The demand for osteosynthesis devices used for the fixation of fractured bones has been increasing yearly, along with the increase in the number of elderly patients with bone fractures, such as osteoporotic fracture. Regarding the evaluation of the mechanical performance of osteosynthesis devices, Arbeitsgemeinschaft für Osteosynthesefragen (AO) in Switzerland states that sufficient strength and stiffness are required to withstand the load that is applied during clinical bone reduction processes. As the metallic materials for osteosynthesis devices, stainless steel, cobalt (Co)-chromium (Cr)-molybdenum (Mo) alloys, commercially pure titanium (C.P. Ti), and Ti alloys, which provide good balance between strength and stiffness, as well as excellent fatigue properties, have been used. The corrosion resistance and biocompatibility of stainless steel (stainless) can be improved by increasing the amount of each of the elements added to the steel, such as Cr and Mo. In addition, fatigue strength can be increased to a level that is equivalent to that of Ti alloys by adding nitrogen (N) and 20% cold working. The fatigue strength of an industrial Ti material can be increased by increasing the amounts of trace elements, such as oxygen (O) and iron (Fe), whereas the fatigue strength of C.P. Ti grade 4 (C.P. Ti G 4) can be made similar to that of Ti alloys by 20% cold working [1]. Ti alloys have higher corrosion resistance and biocompatibility than C.P. Ti, owing to the addition of Mo, zirconium (Zr), niobium (Nb), tantalum (Ta), and so forth [2]. Ti-15Zr-4Nb-4Ta alloy has been developed in Japan as a highly biocompatible alloy for long-term biomedical applications [1,3] and it is standardized in JIS T 7401-4 [4].

The development of orthopedic implants that are optimized for each patient’s needs or skeletal structure (custom-made orthopedic implants) has been made possible, owing to advances in fabrication techniques [5,6,7,8]. Along with the rapid progress of three-dimensional (3D) layer manufacturing technologies in the medical field, the development of orthopedic implants that are customized to the skeletal structure and symptoms of each patient is now possible. To obtain regulatory approval for osteosynthesis devices produced by three-dimensional (3D) layer manufacturing in Japan, evaluation of the fatigue properties of the base metal, and the durability of the osteosynthesis devices is desired [9,10]. To develop products by 3D layer manufacturing with high mechanical reliability, the evaluation of the effects of the implant design and material on their mechanical performance is important. Therefore, we consider it important to examine the relationship among the strength, stiffness, and durability of osteosynthesis devices that are widely used in clinical practice. Reentry, Ids Co., Ltd. (Suita, Osaka, Japan), a dental material supplier, acquired Pharmaceutical Affairs Approval for the use of SP2 Co-Cr-Mo alloy powder as a dental material for 3D layer manufacturing from the Japanese Minister of Health, Labour, and Welfare in April 2018.

The breakage of metallic osteosynthesis devices in clinical use, namely, bone plates [11,12,13], intramedullary nail rods [14,15,16,17,18,19], spinal rods [20,21], compression hip screws (CHSs) [22], short femoral nails [23,24,25,26,27,28,29], and metaphyseal plates [22,30], due to fatigue fracture in the stress concentration region has been reported. In particular, the stress concentration region around a hole leads to crack initiation, resulting in fatigue fracture. The effect of the surface stress distribution on the durability of the orthopedic implant devices has been analyzed by thermoelastic stress measurement [31]. For the mechanical evaluation of osteosynthesis devices, the four-point bending test and the compression bending test are widely performed [32,33,34,35,36,37,38,39]. To evaluate the mechanical stability, finite element analysis is also performed [40,41,42,43]. On the other hand, the four-point bending test of osteosynthesis devices themselves has been recommended for the mechanical evaluation of bone plates, intramedullary nail rods, spinal rods, and screws in ASTM F382 [44], ASTM F1264 [45], and JIS T 0312 [46]. The four-point bending test for bone plates has also been recommended in ISO 9585 [47]. ASTM F382 and JIS T 0312 recommend a (distance between loading rollers):(distance between supporting rollers) ratio of 1:3. Otherwise, ISO 9585 recommends setting the ratio to 1:2. The evaluation of the compression bending stiffness and compression bending strength of CHSs, short femoral nails, and metaphyseal plates is recommended in ASTM F384 [48] and JIS T 0313 [49]. A standard method of statistical fatigue testing has been established in JSME S 002 [50] by the Japan Society of Mechanical Engineers (JSME). A standard evaluation method for the fatigue reliability of metallic materials involving the application of regression to S–N curves (JSMS-SD-06-08) [51] has been provided by the Japan Society for Materials Science (JSMS) Committee on Fatigue and Reliability Engineering. 

In this study, we focused on the development of test jigs and the optimization of test conditions for mechanical tests while using various osteosynthesis devices to obtain mechanical data in accordance with the above international and national standards. Specifically, we examined the relationships between the bending strength and bending stiffness and between the bending strength and the durability of various osteosynthesis devices to clarify the effects of base materials on their properties, such as the relationships between tensile strength and Young’s modulus and between the tensile strength and fatigue strength obtained by room-temperature tensile and fatigue tests of raw materials. One of the purposes of this study is to develop a method of estimating the durability from strength by obtaining relational equations for various osteosynthesis devices. Another purpose is to promote the development of implants that are suitable for the skeletal structures of patients using 3D layer manufacturing technologies. Therefore, we examined the bending strength, bending stiffness, and durability of various types of conventional osteosynthesis devices used as parts that are subjected to loading. Four-point bending tests and compression bending tests of osteosynthesis devices (bone plates, intramedullary nail rods, spinal rods, CHSs, short femoral nails, and metaphyseal plates) were carried out to measure the bending stiffness, bending strength, and durability, in accordance with JIS T 0312 and JIS T 0313. In addition, the room-temperature mechanical properties of miniature specimens cut from each osteosynthesis device and their microstructure were examined. The results that were obtained in this study will be useful for developing highly durable devices using new technologies, such as 3D printing.

## 2. Materials and Methods 

### 2.1. Test Samples

The following osteosynthesis devices used worldwide were examined in this work. Regarding the number of holes in the bone plates, eight holes were mainly used in accordance with ISO 9585, ASTM F382, and JIS T 0312. The osteosynthesis devices that were made of Ti-15Zr-4Nb-4Ta (Ti-Zr) alloy, which are described below, were machined to the same shape as the Ti-6Al-4V (Ti-6-4) alloy devices for comparison among the different Ti alloys. 

(1) Straight plates (mainly eight holes, a few with nine, seven, six holes) manufactured by (i) DePuy Synthes {C.P. Ti G 4; broad and narrow LC-DCPs (426-580, 424-580), condylar plate (95°, 437-960), broad, narrow, and small LC-LCPs [426-581S, 424-581S (partly containing Ti-Zr), 423-581S], LCP reconstructions (445-081S, 429-381S), 1/3 circular small (441-380), and mini-DCP (443-580); lengths, 142, 142, 156, 152, 144, 103, 114, 151, 100, 42; widths, 17.5, 13.5, 16, 17.5, 13.5, 11, 10, 12, 9, 5; thicknesses, 6, 4.6, 5.6, 5.2, 4.6, 3.3, 3.1, 3, 1, 1.5 mm, respectively, and stainless; condylar plate (237-960), 1/3 circular small (241–380), mini-DCP (243–580)}; (ii) Zimmer Biomet [stainless; broad, narrow, and small compressions (00-4945-08-01, 00-4945-008-00, 00-4927-008-00, 00-4935-008-00), 1/3 circular small (00-4935-008-03), 1/4 circular small (0-492-00-03), ECT (0-245-06-08): lengths, 135, 135, 68, 97, 97, 63, 64; widths, 15.5, 12, 8, 9, 9, 7, 6; thicknesses, 4.5, 4, 2.7, 3.5, 1, 1, 1 mm]; (iii) MDM (active compression: Ti-6-4; length, 104; width, 10; thickness, 3.5 mm); (iv) Stryker [narrow compression (620208S): Ti-6-4 or Ti-Zr; length, 146; width, 13; thickness, 4.5 mm]; (v) Mizuho [Ti-6-4 or Ti-Zr; JP broad, narrow and small (01-902-08, 01-902-06, 01-903-6, 01-903-8, 01-903-10, 01-904-06); lengths, 161,125, 125, 156, 192, 77; widths, 17.5, 17.5, 17.5, 13.5, 13.5, 11; thicknesses, 5, 5, 5, 4, 4, 3 mm; holes, 8, 6, 6, 8, 10, 6)]; (vi) Teijin Nakashima Medical (THA; Ti-6-4 or Ti-Zr; lengths, 200, 160; widths, 13, 13; thicknesses, 4, 3.5 mm; holes, 9, 7). 

(2) Intramedullary femoral nail rods manufactured by (i) DePuy Synthes [cannulated (474-041VS, 474-241VS); Ti-6-4; outer diameters, 10 (circular), 12 (hexagonal star); inner diameters, 4, 5.4 mm (circular)]; (ii) Zimmer Biomet [stainless; M/DN retrograde (00-2240-010-30, 00-2240-012-30), GT Femoral (47-2492-401-08, 47-2492-401-09); outer diameters, 10, 12 (both quadrangle star), 8.3, 9.3; inner diameters, 4.8, 6.2, 4.3, 4.7 mm]; (iii) Stryker [Ti-6-4; T2 Femur (1828-1030S,1828-1230S); outer diameters, 10, 12 mm (both quadrangle star); inner diameters, 4.9, 5 mm]; (iv) Smith & Nephew [Trigen (7163-4230, Ti-6-4), Russell Taylor (12-1924, stainless); outer diameters, 10 (circular), 10 (triangle); inner diameters, 5.4 (circular), 3.7 mm (triangle)].

(3) Spinal rods manufactured by (i) Medtronic [Vertex Select: Ti-6-4, machined and blasted Co-28Cr-6Mo (Co-Cr-Mo) rods; diameters, 3.5, 4.75, 6 mm]; (ii) Zimmer Biomet (C.P. Ti G 4; diameter, 5.5 mm); (iii) Robert Reid (Isora: C.P. Ti G 2; diameter, 6.4 mm)]. 

(4) CHSs manufactured by (i) DePuy Synthes [Tubeplate; Ti-6-4 (481-160VS, 481-360VS) and stainless (281-160, 281-360); neck shaft angles, 135, 145°; plate length, 110; plate width, 19; plate thickness, 5.8 mm; holes, 6; barrel length, 38; lag screw lengths, 60, 70, 80, 90, 110 mm]; (ii) Zimmer Biomet [Ti-Versa Fx Ⅱ, Ti-6-4 (48-1201-135-06, 48-1201-135-03) or Ti-Zr: neck shaft angles, 135, 145°; plate lengths, 70, 120; width, 19; thicknesses, 5.8, 8 mm; holes, 3, 6; barrel length, 37; lag screw lengths, 70, 80, 110 mm]; (iii) MDM [ACE (14029); Ti-6-4; neck shaft angle, 135°; plate length, 128; width, 19; thickness, 6 mm; holes, 6; barrel length, 24.5; captured screw lengths, 80, 110 mm]; (iv) Stryker [Omega Plus (3368-1-105, 3368-1-304); Ti-6-4; neck shaft angles, 135, 145°; plate lengths, 95, 117; width, 16.3; thickness, 6.5 mm; holes, 5, 4; barrel length, 31; lag screw lengths, 70, 80, 110 mm]; (v) Teijin Nakashima Medical (KL4; Ti-6-4: neck shaft angles, 135, 145°; plate length, 90; width, 18; thickness, 5 mm; holes, 4; barrel length, 35; lag screw lengths, 70, 80, 110 mm); (vi) Mizuho (01-800-06; Ti-6-4; neck shaft angle, 135°; plate length, 100; width, 18; thickness, 9 mm; holes, 3, 4; barrel length, 38; lag screw length, 70 mm); (vii) Japan Medicalnext (980047-K02609, PWO135-04-2, 980030-K02602; Ti-6-4; neck shaft angle, 135°; plate lengths, 78, 128, 150; widths, 18, 22; thickness, 5 mm; holes, 4, 6, 8; barrel lengths, 32, 38; lag screw lengths, 80, 110, 130 mm). 

(5) Short femoral nails manufactured by (i) Stryker [Gamma 3 (3125-0170S, 3125-2170S, 3130-0170S, 3130-2170S); Ti-6-4; neck shaft angles, 125, 130°, total length, 170; proximal diameter, 15.5; distal diameters, 10, 12; lag screw length, 80 mm]; (ii) Zimmer Biomet [ITST (00-2256-180-10, 00-2256-180-11, 00-2256-180-12); stainless; neck shaft angle, 130°; total length, 180; proximal diameter, 16; distal diameters, 10, 11, 12; lag screw length, 80 mm]; (iii) Smith & Nephew [IMHS (7110-3010, 7166-4411, 7166-4611, 7166-4613); stainless or Ti-6-4; neck shaft angles, 130, 135°; total lengths, 165, 180; proximal diameters, 16.5, 17.5; distal diameters, 10, 11, 13; lag screw length, 80 mm]; (iv) DePuy Synthes [PFN (473-122VS); Ti-6Al-7Nb (Ti-6-7): neck shaft angle, 130°; total length, 200; proximal diameter, 16.5; distal diameter, 11; lag screw lengths, 80, 110 mm].

(6) Epiphyseal plates manufactured by (i) Stryker [T-shaped plate (T-plate, 620418S) and T-shaped buttress plate (T-buttress plate, 620456); Ti-6-4; lengths, 143, 113; width, 16.5; thickness, 2.5 mm; holes, 8, 6]; (ii) DePuy Synthes [T-plate (440-180) and T-buttress plate (440-360); C.P. Ti G 4; lengths, 148, 112; width, 17; thickness, 2.5 mm; holes, 8, 6].

Three each of the bone plates, intramedullary nail rods, spinal rods, CHSs, short femoral nails, and metaphyseal plates were prepared for each bending test. At least six osteosynthesis devices were prepared for the durability test.

### 2.2. Bending Test of Osteosynthesis Devices

An Instron 8874 mechanical testing system was used for each bending test. Four-point bending tests and compression bending tests were conducted on various osteosynthesis devices, in accordance with JIS T 0312 and JIS T 0313, respectively. In particular, we manufactured jigs that fit the osteosynthesis devices with various shapes that were used in this work. The test conditions were controlled using a personal computer. The bending stiffness and bending strength were calculated using Instron mechanical testing software (Bluehill 2). Details of the bending tests of the various osteosynthesis devices are given below. The mean and standard deviation were calculated from the results of bending tests of three specimens of each type of osteosynthesis device.

#### 2.2.1. Four-Point Bending Tests 

The bending stiffness, bending strength, and durability of bone plates, intramedullary nail rods, and spinal rods were measured by the four-point bending test. Figure 1a shows a schematic illustration of the four-point bending test of bone plates, intramedullary nail rods, and spinal rods. Jigs were used for the four-point bending test, which allow for the change in the positions and diameters of the loading and supporting rollers in accordance with the size of the specimens. They also reduce the constraint of bending that is caused by the loading. For eight- or nine-hole bone plates, two loading rollers were set outside the central two holes with the (distance between loading rollers):(distance between supporting rollers) ratio equal to 1:3 (the supporting rollers were placed at the two outermost holes of the loading roller). For six- or seven-hole bone plates, this ratio was 1:2 (the supporting rollers were placed at the outermost hole of the loading roller). Additionally, to examine the effect of the (distance between loading rollers):(distance between supporting rollers) ratio on the four-point bending strength and bending stiffness, supporting rollers were set at the two outermost holes or the outermost hole with the same eight-hole bone plates (h:k = 1:1 or 2:1, as shown below). For the four-point bending tests of intramedullary nail rods and spinal rods, the h:k ratios were 38:114 and 30:60 (mm), respectively. The crosshead speed that was used to measure both the static bending stiffness and the bending strength was 10 mm/min. A bending load was applied to specimens so that they completely curved, and the load–displacement (L–D) curve was measured. The offset displacement was assumed to be 0.2% of the distance between the supporting rollers and the loading rollers. Using the slope of the L–D curve and the offset load that was obtained from the offset displacement, as shown in Figure 1e, we calculated the bending strength (N·m) as M = h × P/2, where P is the offset load (N) and h is the distance between the supporting roller and the loading roller (m). The bending structural stiffness (N·m^2^) was calculated as E = (2h + 3k) × A × h^2^/12, where A is the maximum slope of the linear elastic portion of the L–D curve, h is the distance between the nearest supporting roller and the loading roller (m), and k is the distance between two loading rollers (m). For the four-point bending tests of intramedullary nail rods, the central region of the rod was cut so that the length of the rod was ~130 mm. The four-point bending test of the thus obtained specimen was carried out with the ratio (distance between loading rollers):(distance between supporting rollers) = 1:3 (38:114 mm). For spinal rods, the rod was cut to a length of ~100 mm and the h:k ratio was 30:60 (mm), since the spinal rods had a small diameter. The compression bending strength (M) and bending structural stiffness (E) for intramedullary nail rods and spinal rods were measured using the same calculation formulas, as above.

#### 2.2.2. Compression Bending Tests 

Figure 1b–d show the schematics of the jigs that were used for the compression bending tests of metaphyseal plates, CHSs, and short femoral nails, respectively. The lag screw of a CHS or a short femoral nail was fixed with a screw to the lag screw fixation jig (Figure 1c,d). The angle of the jig changes, so that the distance moved by the lever arm of the lag screw, owing to the loading becomes constant. Auxiliary jigs to accommodate the inside curve of the side plate of each specimen and the positions of holes were developed for CHSs and epiphyseal plates. In addition, an auxiliary jig to tuck and fix the nail region was developed for short femoral nails. In the test using CHSs and short femoral nails, a load was applied to the center of the ridge of a lag screw. The compression bending stiffness and compression bending strength were measured while the topmost hole of the side plate of the CHS was not fixed (open), in accordance with JIS T 0313. Short femoral nails were fixed at a position 60–75 mm from the distal end of the nail, depending on the length and shape of the nail. Epiphyseal plates were fixed to the jig using the second and later holes, while the topmost hole of the side plate was not fixed (open), and a load was applied to the upper end of the plate during the test. Bearings were attached to the lower part of the jig, so as not to disturb the deformation that is caused by the loading. In these static compression bending tests, the L–D curves were measured, while a bending load was applied to specimens at a crosshead speed of 10 mm/min until they were completely bent. The compression bending stiffness (E) was defined as the slope (N/m) of the L–D curve. The compression bending strength (M) for each sample was defined as M = L × P (N·m), where L is the distance from the side plate or the inner side of the nail to the load point (lever arm (m), as shown in Figure 1c,d, respectively) and P is the offset load (N). The offset load corresponding to 0.2% offset of the displacement L was determined from the L–D curve measured in each compression bending test. Subsequently, the compression bending strength was calculated using the offset load. The compression bending stiffness (E) and bending strength (M) of epiphyseal plates were measured by the same method.

### 2.3. Durability Tests of Osteosynthesis Devices

Five Instron 8874 mechanical testing systems were used for each durability test. The test conditions were controlled using the Instron software (Max), and changes in the waveform during the test were constantly monitored. M–N curves (maximum bending moment vs number of cycles to failure on logarithmic scale) were measured for the four-point bending durability test and the compression bending durability test. The maximum bending moments (M) were calculated as M = h × Pm/2 for the four-point bending durability test and M = L × Pm for the compression bending durability test, where Pm is the maximum load (N) in the four-point bending durability test or compression bending durability test. 

Using the four-point bending test jigs, the durability of bone plates, intramedullary nail rods, and spinal rods was evaluated under the following conditions: (distance between loading rollers):(distance between supporting rollers) ratio = 1:3, except for spinal rods, h:k ratio = 1:2 for spinal rods, sinusoidal waves, and load ratio R (minimum load/maximum load) = 0.1. For bone plates, sinusoidal waves with a frequency in the range of 1–8 Hz (mainly 5 Hz) were applied to examine the effect of the frequency. For intramedullary nail rods and spinal rods, sinusoidal waves with a frequency of 3 Hz were used. From the obtained M–N curves, the durability limit, which is the maximum moment corresponding to 1 × 10^6^ cycles, was determined.

A compression bending durability test was carried out using similar jigs to evaluate the static compression. Sinusoidal waves with frequencies of 3 or 5 Hz (CHSs), 3 Hz (short femoral nails), and 1 Hz (epiphyseal plates) were applied at a stress ratio R = 0.1 for more than 1 × 10^6^ cycles. From the obtained M–N curve, the durability limit after 1 × 10^6^ cycles was determined by the same method. 

### 2.4. Room-Temperature Tensile Test and Microstructural Observation

Three uniform rod specimens (gage length, 7.5 mm; diameter, 1.5 mm; or gage length, 15 mm; diameter, 3 mm) were cut from each bone plate, intramedullary nail rod, spinal rod, CHS, short femoral nail, and metaphyseal plate, in accordance with the shape of the specimen. The strain rate was set to 0.5%/min until failure. We analyzed the cross-sectional microstructure by optical microscopy and transmission electron microscopy (TEM). Nitric hydrofluoric acid solution (3%) was used as an etching agent. The 3-mm-diameter disc specimens were electropolished with 95% methanol +5% perchloric acid solution.

### 2.5. Statistical Analysis

We calculated the correlation coefficients (r) between the bending stiffness and the bending strength and between the durability limit and the bending strength for each bone plate, intramedullary nail rod, spinal rod, CHS, short femoral nail, and metaphyseal plate. Additionally, linear regression analysis was performed between the bending stiffness and the bending strength, and between the durability limit and the bending strength, in both cases assuming a line passing through the origin. Moreover, the M–N curve, the durability limit of various osteosynthesis devices, and its standard deviation were calculated with statistical analysis software that is based on JSMS-SD-06-08 [51].

## 3. Experimental Results and Discussion

### 3.1. Bending Stiffness and Bending Strength of Osteosynthesis Devices

Because the recommended h:k ratios in ISO 9585, ASTM F382, and JIS T 0312 are different, we examined the effect of the h: k ratio. Figure 2a,b show the effect of the h:k ratio on the four-point bending structural stiffness and bending strength, respectively. The error bars represent the standard deviation. The straight lines were obtained by linear regression. As shown in Figure 2a, the effect of the h:k ratio on the four-point bending structural stiffness was small. On the other hand, the bending strengths that were measured at h:k = 1:1 were lower than (about 82% of) those measured at h:k = 2:1 (coefficient of determination, R^2^ = 0.99). From these results, it was clarified that the h:k ratio affects the four-point bending structural stiffness and bending strength, but in many tests, h:k = 1:1 recommended by ASTM F382 and JIS T 0312 is employed. Figure 3 shows the relationship between the bending structural stiffness and bending strength of (a), (b) bone plates, (c) intramedullary nail rods, and (d) spinal rods that were obtained by the four-point bending test. In Figure 3c, the typical cross-sectional shapes (e.g., circle, star, and triangle) of the intramedullary nail rods, which affected the bending structural stiffness and bending strength, are shown. The correlation coefficients (r) between the bending stiffness and bending strength (also, given in Figure 3) are 0.86 to 0.96, which indicate a strong correlation. The bending structural stiffnesses of the bone plates, intramedullary nail rods, and spinal rods almost linearly increase with increasing four-point bending strength. The slopes of the straight lines [= (bending structural stiffness)/(bending strength)] for the bone plates, intramedullary nail rods, and spinal rods are around 0.3 (R^2^ = 0.73–0.92), which were measured under a condition of h:k = 1:1. Regarding the relationship between the four-point bending structural stiffness and the bending strength, we obtained [bending structural stiffness (N·m^2^)] = 0.32 × [four-point bending strength (N·m)] (R^2^ = 0.88) by regression. The relationship for the highly biocompatible Ti-Zr alloy was also linear. The four-point bending strength is expressed as
M = 1/2 × P × h = σ × Z = α × σ × Z.(1)

Here, *σ* is the surface stress that is generated by four-point bending, α is the stress concentration factor, and *Z* is the section modulus of the osteosynthesis device, which depends on the cross-sectional shape and affects the bending structural stiffness and bending strength [31]. The bending structural stiffness and bending strength increase with increasing *Z*.

Figure 4 shows the relationship between the compression bending stiffness and compression bending strength of (a), (b) CHSs, (c) short femoral nails, and (d) epiphyseal plates. The same marks in Figure 4a,b indicate the results for the same specimen. The correlation coefficient (r) for the epiphyseal plates is 0.91. Since the data for CHSs and short femoral nails are scattered, r is not given. In Figure 4a,b, the effects of the lag screw length (70, 80, and 110 mm) and the neck shaft angle (135 and 145°) on the compression bending stiffness and compression bending strength are shown. The compression bending strength and compression bending stiffness at a neck shaft angle of 145° tended to be higher than those at a neck shaft angle of 135°. With increasing the compression bending strength, the compression bending stiffness tends to linearly increase depending on the features of the side plate (e.g., width, thickness, and barrel length). However, with an increasing lag screw length (60, 70, 80, and 110 mm), the compression bending stiffness tends to decrease from 0.21 to 0.06 MN·m^−1^ (Tubeplate, 481-160VS). The compression bending stiffness and bending strength of stainless specimens are slightly higher than those of Ti alloy specimens, because of the higher stiffness of stainless steel. Figure 4c shows the result of the compression bending test of short femoral nails. With increasing compression bending strength, the compression bending stiffness linearly increases, depending on the proximal diameter, proximal thickness, and the shape of the holes of the specimens. The compression bending strength and bending stiffness of short femoral nails having the same proximal diameter also increased with increasing distal diameter, as shown in Figure 4c. The effect of the proximal diameter of nails on the results of the compression bending test is significant. Figure 4d shows the relationship between the compression bending stiffness and compression bending strength of T-plates and T-buttress plates. The compression bending stiffness and compression bending strength of the T-plates were higher than those of the T-buttress plates. 

### 3.2. Durability of Osteosynthesis Devices

Figure 5 and Figure 6 show the M–N curves that were obtained by the four-point bending test. The arrows on the symbols indicate the specimens for which fracture did not occur. The numbers shown in Figure 5 correspond to the numbers of the osteosynthesis devices that are listed in Section 2.1. In Figure 5, the maximum moments of a stainless (narrow compression, 00-495-008-00) bone plate and three bone plates made of different Ti alloys (C.P. Ti G 4, Ti-6-4, and Ti-Zr) obtained by the four-point bending test using sinusoidal waves with a frequency of 5 Hz in air are shown. The effect of the frequency on the durability limit of LC-LCP 424-581S was small, being in the frequency range of 1–8 Hz. A durability limit of 10^6^ cycles corresponds to one year of clinical use [31]. The M–N curves of bone plates made of 20% cold-worked C.P. Ti G 4 and highly biocompatible Ti-Zr alloy were similar when their shapes were similar. The maximum moment of a Stryker-type bone plate (620208) that was made of Ti-6-4 was compared with that made of Ti-Z alloy of the same shape, as shown in Figure 5b, and it was found that the difference was attributable to the difference in the manufacturing processes of die forging and annealing. The durability of intramedullary nail rods, as shown in Figure 6a, was found to be affected by the cross-sectional shape (section modulus). For spinal rods (Figure 6b), the effect of the round bar diameter on the durability was relatively large. 

Figure 7a shows the M-N curves of six CHSs (five Ti alloy specimens and one stainless specimen), having the same lag screw length of 80 mm. The fatigue characteristics of sliding-type specimens tended to deteriorate, owing to the deformation of the rail groove on the plate side. The position of failure during the durability test moves from the topmost screw hole of the side plate to the lag screw region with decreasing maximum moment. Figure 7b shows the M–N curves of short femoral nails with a lag screw length of 80 mm. The durability limit is the lowest for the specimen with a neck shaft angle of 125° and a small proximal diameter of 15.5 mm. Figure 7c shows the M–N curves of the T-shaped plates and T-shaped buttress plates. The durability limit of the T-buttress plates tended to be lower than that of the T-plates.

### 3.3. Relationship Between Durability Limit and Bending Strength

Figure 8 shows the relationship between the durability limit (at 10^6^ cycles) and the bending strength plotted on a log–log graph. The durability limit of various types of osteosynthesis devices linearly increased with increasing bending strength (r = 0.93). The slopes of the bending stiffness–bending strength graphs for bone plates, spinal rods, intramedullary nail rods, CHSs, short femoral nails, and epiphyseal plates are 0.66 (R^2^ = 0.94), 1.05 (R^2^ = 0.96), 0.65 (R^2^ = 0.31), 0.92 (R^2^ = 0.11), 0.89 (R^2^ = 0.51), and 1.26 (R^2^ = 0.96), respectively. As explained above, osteosynthesis devices can be roughly divided into two groups: those with a durability limit of approximately 0.7 times the four-point bending strength and those with a durability limit that is equivalent to the compression bending strength. The relationship (durability limit at 10^6^ cycles) = 0.67 × (bending strength) (N·m) (R^2^ = 0.85) was obtained for bone plates, spinal rods, intramedullary nail rods, CHSs, short femoral nails, and epiphyseal plates by regression. The relationship (durability limit at 10^6^ cycles) = 0.66 × (four-point bending strength) (N·m) (R^2^ = 0.86) was obtained for bone plates, spinal rods, and intramedullary nail rods. The relationship (durability limit at 10^6^ cycles) = 0.91 × (compression bending strength) (N·m) (R^2^ = 0.89) was obtained for CHSs, short femoral nails, and epiphyseal plates. Additionally, this finding corresponds to the fact that the fatigue strengths of 20% cold-worked stainless, forged Co-Cr-Mo, 20% cold-worked C.P. Ti G 4, Ti-6-4, and Ti-Zr alloy (annealed) specimens are approximately 0.79, 0.74, 0.69, 0.73, and 0.80 times the tensile strength of each specimen, respectively [3,52]. 

Table 1 shows the tensile properties of each material obtained from specimens cut from various osteosynthesis devices that were made by different manufacturers. The means of the 0.2% proof strength (σ_0.2%PS_), ultimate tensile strength (σ_UTS_), total elongation (T.E.), and reduction in area (R.A.), together with their standard deviations, were calculated for three test specimens of each type of the osteosynthesis device. The mechanical strength and ductility of these osteosynthesis devices were excellent. Figure 9 shows optical micrographs and TEM images of the osteosynthesis materials. The 20% cold-worked stainless, 20% cold-worked C.P. Ti G 4, and annealed Ti-6-4 and Ti-Zr alloys consist of fine structures, which are considered to be related to the excellent durability that is shown in Figure 8.

Thus, the relationship between the bending strength and durability of osteosynthesis devices was clarified. It is possible to predict the durability from the bending strength of osteosynthesis devices using these regression equations. The results that wereobtained in this study are considered to be useful for developing highly durable devices, the development of implants that are suitable for the skeletal structure of patients using 3D layer manufacturing technologies, and for the approval of commercial medical devices.

## 4. Conclusions

We examined the bending strength, bending stiffness, and durability of various types of osteosynthesis devices (bone plates, intramedullary nail rods, spinal rods, CHSs, short femoral nails, and metaphyseal plates) to promote the development of implants that are suitable for the skeletal structure of patients using 3D layer manufacturing technologies. Four-point bending tests and compression bending tests of osteosynthesis devices were carried out to measure the bending stiffness, bending strength, and durability. 

1. The bending stiffness of bone plates, spinal rods, and intramedullary nail rods almost linearly increased with increasing four-point bending strength. The slopes of the straight lines [= (bending stiffness)/(bending strength)] for bone plates, spinal rods, and intramedullary nail rods were approximately 0.3 (R^2^ = 0.73–0.85).2. With increasing the compression bending strength, the compression bending stiffness of CHSs, short femoral nails, and metaphyseal plates tended to increase linearly, depending on the cross-sectional shape.3. The durability limit of various types of osteosynthesis devices linearly increased with increasing bending strength (r = 0.93). The relationship (durability limit at 10^6^ cycles) = 0.67 × (bending strength) (N·m) (R^2^ = 0.85) was obtained by regression. This slope of 0.67 was close to the ratio of the fatigue strength to the tensile strength of the raw metals. The relationship for the highly biocompatible Ti-15Zr-4Nb-4Ta alloy was also linear.4. The mechanical strength and ductility of specimens that were cut from various osteosynthesis devices were excellent and their microstructure consisted of fine structures, which are considered to be related to the excellent durability.

## Figures and Tables

**Figure 1 materials-12-00436-f001:**
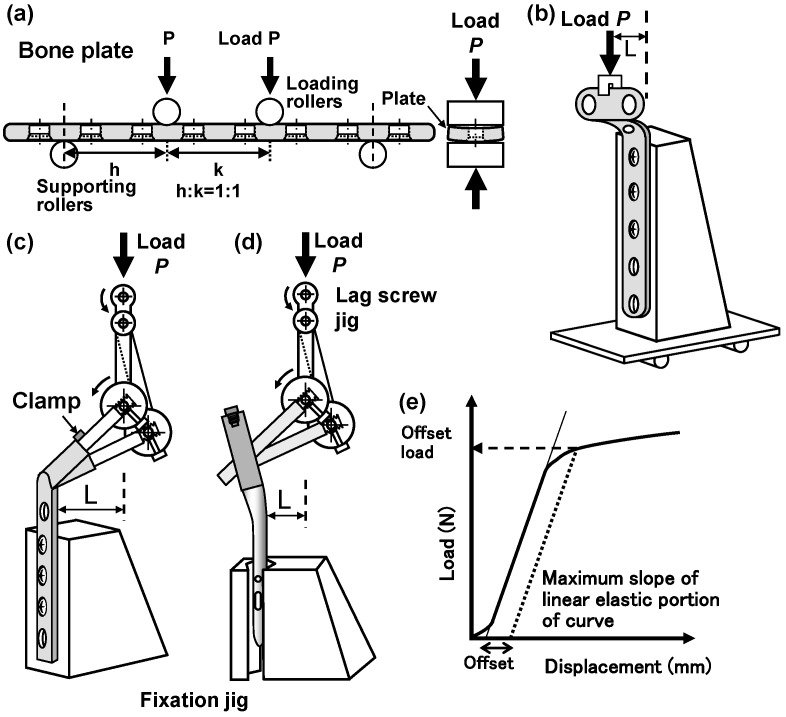
Schematic illustration of (**a**) four-point bending test of bone plates, intramedullary nails, and spinal rods, and (**b**–**d**) jigs for the compression bending test of metaphyseal plates, compression hip screws (CHSs), and short femoral nails, respectively. (**e**) Load versus displacement (L–D) curve for determining the bending properties from four-point bending test or compression bending test results.

**Figure 2 materials-12-00436-f002:**
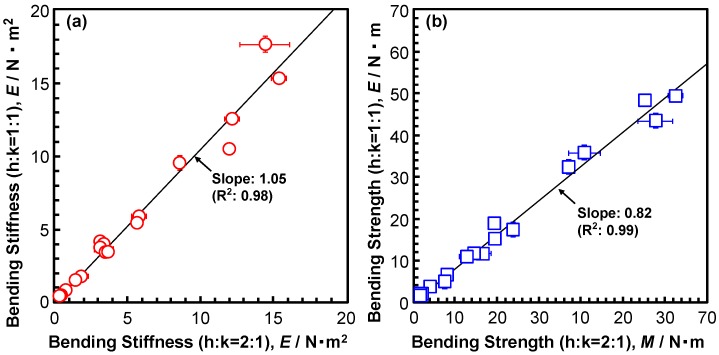
Effect of h:k ratios (h:k = 1:1 or h:k = 2:1) on (**a**) four-point bending stiffness and (**b**) bending strength obtained from four-point bending tests with the same eight-hole bone plates.

**Figure 3 materials-12-00436-f003:**
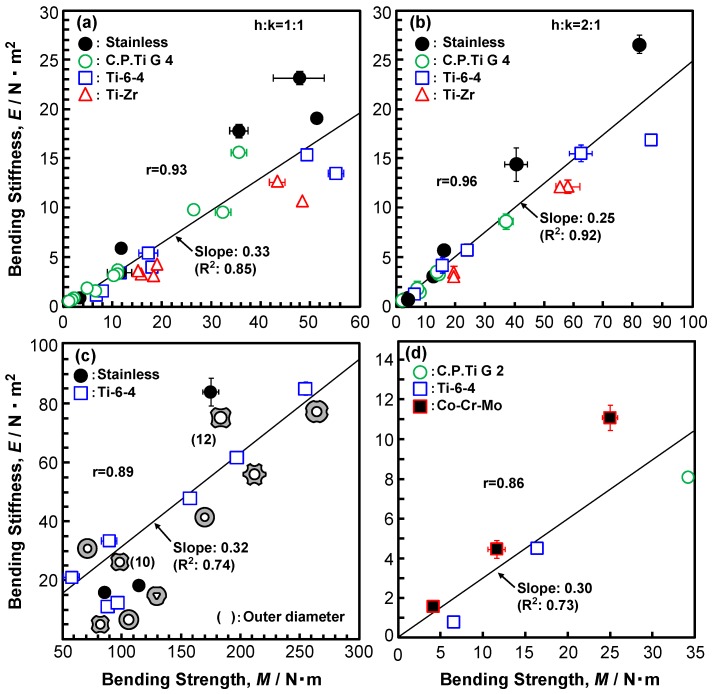
Relationship between bending stiffness and bending strength obtained from four-point bending tests. (**a**) bone plates (h:k = 1:1), (**b**) Bone plates (h:k = 2:1), (**c**) intramedullary nail rods, and (**d**) spinal rods.

**Figure 4 materials-12-00436-f004:**
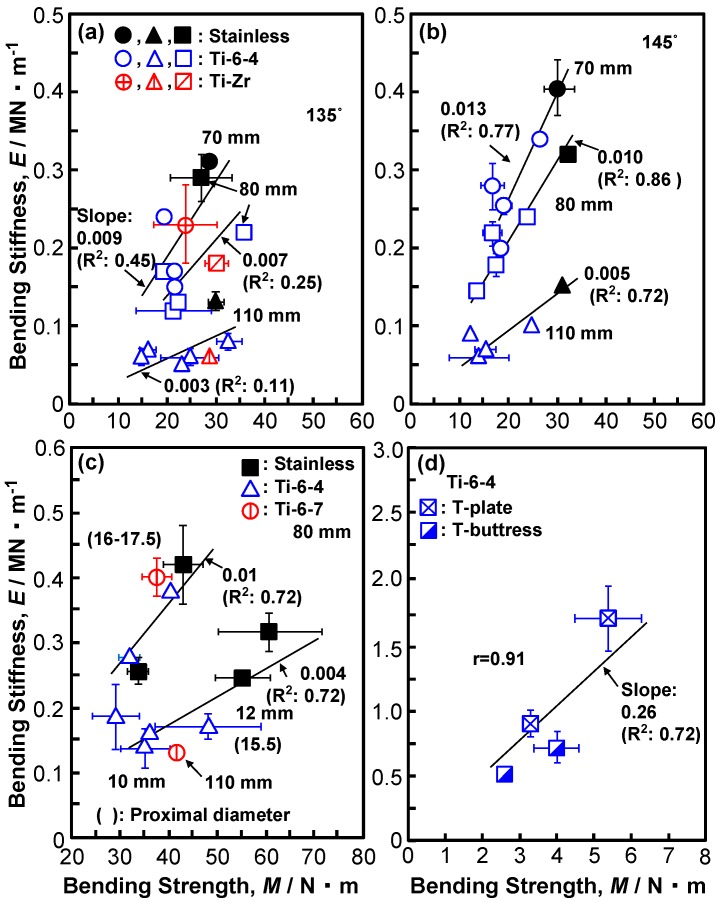
Relationship between bending stiffness and bending strength obtained from compression bending tests. (**a**,**b**) CHSs, (**c**) short femoral nails, and (**d**) epiphyseal plates.

**Figure 5 materials-12-00436-f005:**
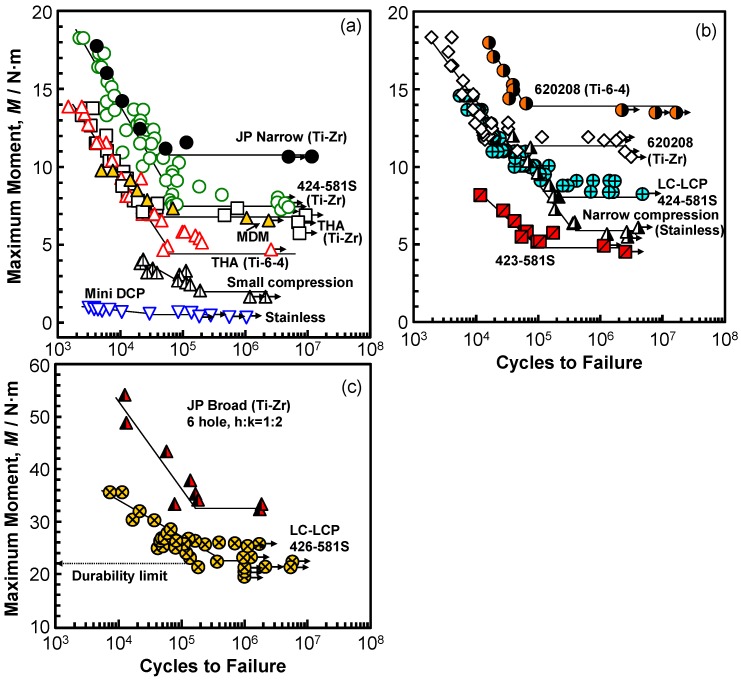
M–N curves of various bone plates obtained by four-point bending durability tests. (**a**) Relative small loaded bone plates; (**b**,**c**) relative large loaded bone plates.

**Figure 6 materials-12-00436-f006:**
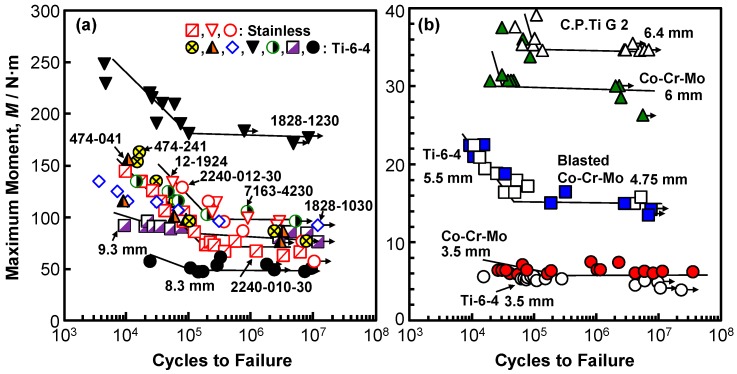
M–N curves of various (**a**) intramedullary nail rods and (**b**) spinal rods obtained by four-point bending durability tests.

**Figure 7 materials-12-00436-f007:**
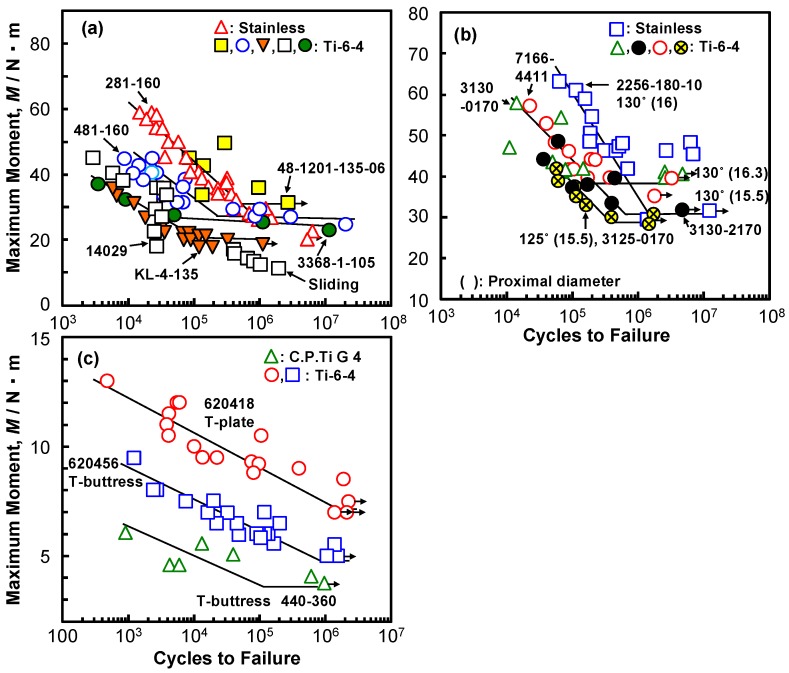
M–N curves of (**a**) CHSs, (**b**) short femoral nails, and (**c**) epiphyseal plates that were obtained by compression bending durability tests.

**Figure 8 materials-12-00436-f008:**
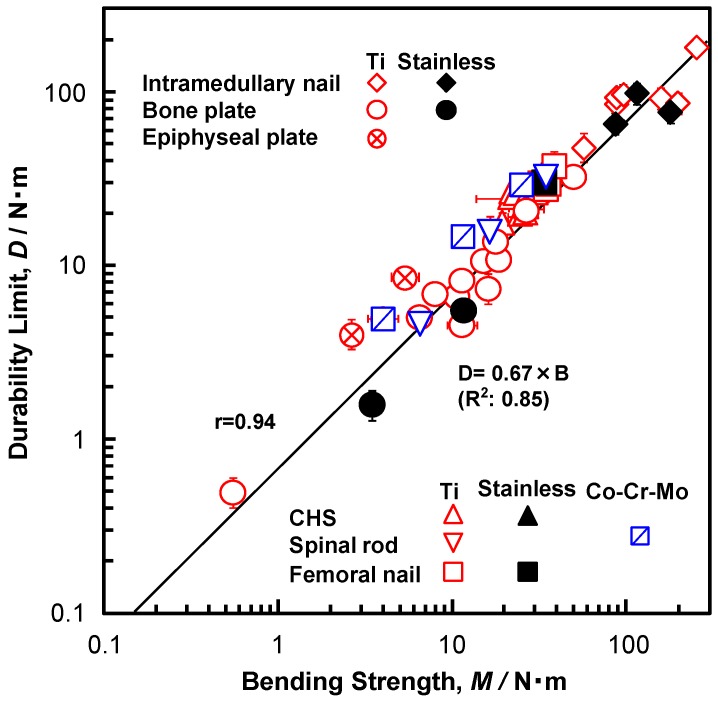
Relationship between durability limit and bending strength of osteosynthesis devices, as determined by four-point bending tests and compression bending tests.

**Figure 9 materials-12-00436-f009:**
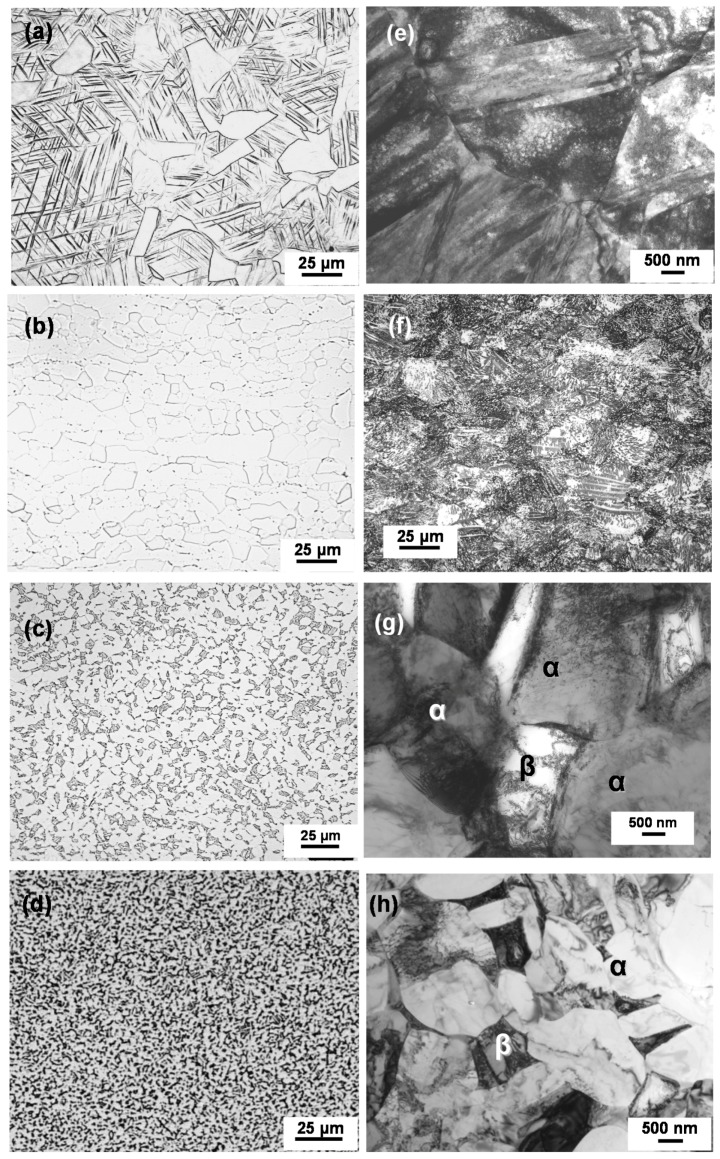
(**a**–**d**) Optical micrographs and (**e**–**h**) transmission electron microscopy (TEM) images of (**a**,**e**) 20% cold-worked stainless, (**b**,**f**) 20% cold-worked C.P. Ti G 4, (**c**,**g**) Ti-6Al-4V alloy, and (**d**,**h**) Ti-15Zr-4Nb-4Ta alloy.

**Table 1 materials-12-00436-t001:** Tensile properties of various osteosynthesis devices.

Alloy	σ_0.2%__PS_ /MPa	σ_UTS_/MPa	T.E.(%)	R.A.(%)
Bone plate				
C.P. Ti G 4	650 ± 13	700 ± 7	21 ± 2	56 ± 3
Ti-6-4	873 ± 26	949 ± 6	24 ± 2	34 ± 7
Stainless	877 ± 17	994 ± 14	21 ± 1	81 ± 3
Ti-Zr	848 ± 2	915 ± 3	21 ± 2	55 ± 3
Spinal rod				
C.P. Ti G 2	322 ± 16	431 ± 9	35 ± 3	68 ± 1
C.P. Ti G 4	597 ± 1	755 ± 1	34 ± 2	54 ± 1
Ti-6-4	848 ± 4	1003 ± 3	20 ± 2	46 ± 4
Co-Cr-Mo	846 ± 10	1315 ± 9	36 ± 2	27 ± 1
Intramedullary nail				
Ti-6-4	874 ± 50	985 ± 3	19 ± 1	34 ± 3
Stainless	790 ± 54	1100 ± 17	26 ± 4	66 ± 3
CHS				
Ti-6-4	917 ± 35	966 ± 31	19 ± 2	45 ± 8
Stainless	971 ± 48	1009 ± 63	15 ± 2	72 ± 5
Short femoral nail				
Ti-6-4	873 ± 79	973 ± 9	20 ± 2	36 ± 4
Ti-6-7	862 ± 18	952 ± 8	21 ± 1	36 ± 4
Stainless	864 ± 109	1101 ± 57	21 ± 1	78 ± 2
High-N stainless	1229 ± 8	1300 ± 5	14 ± 1	33 ± 2
Epiphyseal plate				
C.P. Ti G 4	456 ± 32	719 ± 4	27 ± 2	51 ± 1
Ti-6-4	967 ± 17	1066 ± 12	16 ± 1	42 ± 6

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
