# Peer review of "Strength–Durability Correlation of Osteosynthesis Devices Made by 3D Layer Manufacturing"

_materials, 2019, doi:10.3390/ma12030436_

Round 1
Reviewer 1 Report
This paper examines the bending strength, bending stiffness, and durability of various types of osteosynthesis devices. The testing is complete, and the analysis is solid. I would recommend to accept this manuscript for publication after a few minor revisions.
1. It would be better if the author can provide some more literature evidence, especially in the introduction.
2. Fig 3 and 4 still should highlight the key difference in (a) and (b). (for example, (a) Bone plate (h:k =1:1), (b) Bone plate (h:k =2:1).
3. In Fig 4 and Fig 8, The Bending Strength is sometimes noted as M and sometimes as B. Should them all be M, since M is defined in Equation 1 in Line 255? Also I find it confusing with the expression B / N*m. Is it better to say B (N*m) or B [N/m]?
4. The legends should also be included in Fig. 7 (b) and (c). In Fig. 1, it is also better to include the legend to remind readers which material is tested.
Author Response
Response to Reviewers
Professor Maryam Tabrizian Editor-in-Chief
Materials
Dear Professor Tabrizian:
We highly appreciate the editor and reviewers for their thorough review and for the helpful and insightful comments. On the basis of the comments made by the reviewers, we have made the following major changes to the manuscript. Corrected parts of the manuscript are indicated in red. In particular, we have clarified the purpose of this study in the introduction. Also, since we tested many osteosynthesis devices, we have modified the descriptions of the test methods and results to simplify them. Furthermore, we have emphasized the importance of the test results and the application of regression analysis in this study. We have added co-authors and the abbreviations of symbols used in this work.
The responses to the comments from the three reviewers are given below.
Reviewer 1
On the basis of the comments made by the reviewer, we have made the following changes to the manuscript. Corrected parts of the manuscript are indicated in red.
1. We have added literature that clarifies the purpose of this research.
2. We have added h: k ratios in Figs. 2 and 3.
3. The bending moments in Figs. 4 and 8 have been corrected to M.
4. We have added legends for the tested materials in Figs. 7(b) and 7(c).
Reviewer 2 Report
The manuscript aims at presenting an extensive analysis of various orthopaedic osteosynthesis devices.
Despite, the manuscript is very complicated to read because it does not become sufficiently clear which elements are novel analyses and which elements are references to the literature. As the aims are not formulated precisely, assessing which analyses have been performed and why does not become clear. It is also very hard to derive why specific methods have been used, e.g. different h:k ratios. Have both ratios been used for all the investigations? Maybe tables or a graph detailing all the materials investigated (and methods used to assess their biomechanical properties) would be helpful.
The results are mixed up with the discussion which is not in line with the author guidelines. The order of the sections is not in line with the guidelines as well. The figures are very hard to interpret as a lot of information is missing, so one does not know what all the different signs represent.
I am wondering that one author solely presents such an extensive analysis - were there no group members to support the investigation?
Taking this together, the manuscript required extensive editing to become more clear and precise prior to publication.
Author Response
Response to Reviewers
Professor Maryam Tabrizian Editor-in-Chief
Materials
Dear Professor Tabrizian:
We highly appreciate the editor and reviewers for their thorough review and for the helpful and insightful comments. On the basis of the comments made by the reviewers, we have made the following major changes to the manuscript. Corrected parts of the manuscript are indicated in red. In particular, we have clarified the purpose of this study in the introduction. Also, since we tested many osteosynthesis devices, we have modified the descriptions of the test methods and results to simplify them. Furthermore, we have emphasized the importance of the test results and the application of regression analysis in this study. We have added co-authors and the abbreviations of symbols used in this work.
The responses to the comments from the three reviewers are given below.
Reviewer 2
On the basis of the comments made by the reviewer, we have made the following changes to the manuscript. Corrected parts of the manuscript are indicated in red.
1. We checked all of the manuscript carefully. In particular, we have cited literature to clarify the purpose of this research in the introduction, and as suggested by the reviewer, we have modified the test method, test results, and figure captions. We have given the reason why the h: k ratio was changed in the test method in the first sentence of the experimental results. The numbers shown in Fig. 5 correspond to the numbers of the osteosynthesis devices listed in section 2.1. We have added material information to explain the marks in Fig. 7. We have summarized the importance of the results and the application of regression analysis in this study at end of the results and discussion.
2. It is stated in the author guidelines that the results and discussion can be presented together, and we judge that the results and discussion are easier to understand when they are presented together. We have striven to describe the data as compactly as possible to show how the data changes with the osteosynthesis device design and raw materials. Therefore, the static and dynamic mechanical data of the various osteosynthesis devices are shown separately for the compression bending test and four-point bending test.
3. In this research, it took a very long time to develop the test jig and test conditions for the many types of osteosynthesis devices and to summarize the many test results. We have added coauthors. Finally, to explain the symbols used in this study, abbreviations have been added at the end of the manuscript.
Reviewer 3 Report
Dear authors, I wish to appreciate you for your work. The work is very interesting.
The paper is well written, in general, and deserves publication. Only one comment should be made:
In the Materials and Method you can provide in greater detail information about 3D layer manufacturing parameters.
Author Response
Response to Reviewers
Professor Maryam Tabrizian Editor-in-Chief
Materials
Dear Professor Tabrizian:
We highly appreciate the editor and reviewers for their thorough review and for the helpful and insightful comments. On the basis of the comments made by the reviewers, we have made the following major changes to the manuscript. Corrected parts of the manuscript are indicated in red. In particular, we have clarified the purpose of this study in the introduction. Also, since we tested many osteosynthesis devices, we have modified the descriptions of the test methods and results to simplify them. Furthermore, we have emphasized the importance of the test results and the application of regression analysis in this study. We have added co-authors and the abbreviations of symbols used in this work.
The responses to the comments from the three reviewers are given below.
Reviewer 3
On the basis of the comments made by the reviewer, we have made the following changes to the manuscript. Corrected parts of the manuscript are indicated in red.
It was recommended that we describe the 3D layer manufacturing parameters in Materials and Methods. Since the purpose of this study is to examine the relationship between the bending strength and durability of osteosynthesis devices, we would like to describe then in another paper. Instead, we have described recent trends of 3D layer manufacturing materials in Japan in the introduction.
Round 2
Reviewer 2 Report
The manuscript improved significantly compared to the previous version and all my concerns have been resolved. Thus I recommend to accept this manuscript for publication.
Author Response
We highly appreciate the reviewer for their thorough review.
Thank you very much for your consideration.